# Mapping of mitogen and metabolic sensitivity in organoids defines requirements for human hepatocyte growth

Delilah Hendriks [1,2,3,7] ✉, Benedetta Artegiani [3,7] ✉, Thanasis Margaritis [3], Iris Zoutendijk [3], Susana Chuva de Sousa Lopes[4] & Hans Clevers [1,2,3,5,6] ✉

Mechanisms underlying human hepatocyte growth in development and regeneration are incompletely understood. In vitro, human fetal hepatocytes (FH) can be robustly grown as organoids, while adult primary human hepatocyte (PHH) organoids remain difficult to expand, suggesting different growth requirements between fetal and adult hepatocytes. Here, we characterize hepatocyte organoid outgrowth using temporal transcriptomic and phenotypic approaches. FHs initiate reciprocal transcriptional programs involving increased proliferation and repressed lipid metabolism upon initiation of organoid growth. We exploit these insights to design maturation conditions for FH organoids, resulting in acquisition of mature hepatocyte morphological traits and increased expression of functional markers. During PHH organoid outgrowth in the same culture condition as for FHs, the adult transcriptomes initially mimic the fetal transcriptomic signatures, but PHHs rapidly acquire disbalanced proliferation-lipid metabolism dynamics, resulting in steatosis and halted organoid growth. IL6 supplementation, as emerged from the fetal dataset, and simultaneous activation of the metabolic regulator FXR, prevents steatosis and promotes PHH proliferation, resulting in improved expansion of the derived organoids. Single-cell RNA sequencing analyses reveal preservation of their fetal and adult hepatocyte identities in the respective organoid cultures. Our findings uncover mitogen requirements and metabolic differences determining proliferation of hepatocytes changing from development to adulthood.

Fundamental knowledge surrounding liver function in health and disease has greatly benefitted from rodent models, as they present versatile and tractable experimental systems and can be subjected to genetic perturbation. Liver regeneration has been amply studied through partial hepatectomy experiments[1,2]. Likewise, intricate details of developmental processes underlying embryonic liver growth and maturation have been derived from genetic tracing models[3,4]. It remains less known how these rodent principles relate to liver growth and regeneration in humans. A better understanding of mechanisms and requirements of human hepatocyte growth could help optimizing

[1]Hubrecht Institute, Royal Netherlands Academy of Arts and Sciences, Utrecht, The Netherlands. [2]Oncode Institute, Utrecht, The Netherlands. [3]The Princess Maxima Center for Pediatric Oncology, Utrecht, The Netherlands. [4]Department of Anatomy and Embryology, Leiden University Medical Center, Leiden, The Netherlands. [5]University Medical Center Utrecht, Utrecht, The Netherlands. [6]Present address: Pharma Research and Early Development (pRED) of F. Hoffmann-La Roche Ltd, Basel, Switzerland. [7]These authors contributed equally: Delilah Hendriks, Benedetta Artegiani. ✉e-mail: d.hendriks@hubrecht.eu; b.a.artegiani@prinsesmaximacentrum.nl; h.clevers@hubrecht.eu

therapeutic regenerative strategies for chronic liver diseases. Recent single-cell RNA sequencing studies on fetal and adult human livers highlighted high tissue complexity, cellular heterogeneity, and described novel subpopulations (e.g. refs. [5]–[17]). Follow-on research benefits from experimentation using functional in vitro models in which processes, such as cellular growth, can be live-traced.

In vitro organoid cultures constitute model systems in which processes, such as cellular growth and differentiation/maturation of epithelial cells, can be studied[18]. Tissue-derived organoids have been previously established from both epithelial liver cell lineages[19], i.e. hepatocytes and cholangiocytes (bile duct cells), as separate cultures and each with a specific culture medium. Mouse and human EpCAM⁺ liver cells can give rise to cholangiocyte organoids, which grow in an expansion medium rich in growth factors and Wnt signals and includes, amongst others, the cAMP activator forskolin[20], and for human cells also the TGFβ inhibitor A83-01[21,22]. Different culture conditions to grow hepatocyte organoids from mouse ALB⁺ hepatocytes[23,24] and human AFP⁺ fetal hepatocytes (FH)[23,25,26] were more recently established. The expansion medium for hepatocyte organoids differs by multiple factors, including, amongst others, the lack of forskolin and the essential inclusion of the GSK3β inhibitor CHIR-99021[23]. The growth of adult primary human hepatocytes (PHH) as organoids has remained challenging and resulted in short-living cultures[23]. Thus, essential differences may exist to promote organoid growth of fetal versus adult hepatocytes.

Here, we perform time-resolved transcriptomic and phenotypic characterizations of organoid growth initiated from FHs and PHHs. We uncover specific mitogen and metabolic requirements differences between fetal and adult hepatocytes. Building on these findings, we discover factors boosting FH organoid growth, establish maturation conditions for FH organoids, and design a culture medium promoting improved growth of PHH organoids. Single-cell RNA sequencing of expanding fetal and adult organoid cultures uncovers their cellular identities.

## Results

### Time-resolved transcriptomic signatures of hepatocyte organoid establishment from human fetal liver

To temporally characterize the transcriptomic responses upon hepatocyte organoid outgrowth from human fetal liver, we processed tissues from 2 donors and seeded the resulting liver cell suspension into domes of basement membrane extract (BME) overlaid with hepatocyte-specific expansion medium. We collected tissue fragments and the cell suspension generated thereof, as well as the cultured cells and emerging fetal hepatocyte (FH) organoids at different time points post seeding. We also collected organoids 1 day after passaging of the established FH organoid lines (Fig. 1a). Within 3 days post seeding, small organoids appeared, which continued to expand into typical FH organoid lines[23,25] (Fig. 1b). Immunofluorescence characterization of these lines confirmed their fetal hepatocyte identity (AFP⁺, ALB⁺), while cholangiocyte (bile duct) markers CK7 and CK19 were absent (Fig. 1c). The organoids were broadly Ki-67⁺, displayed occasional polyploid cells, and possessed MRP2⁺ bile canaliculi (Fig. 1d).

We further surveyed the identity of the cells present in an established FH organoid culture from one donor through single-cell RNA sequencing, recovering 1421 cells (Fig. 2a, Supplementary Fig. 1a, b). These analyses confirmed their uniform hepatocyte identity, underscored by e.g. broad *ALB, AFP, TTR, SERPINA1, ASGR1* expression, as well as various other fetal/mature hepatocyte markers (Fig. 2b, Supplementary Fig. 1c). Within these, we noted hepatocytes presenting a more glucose/lipid metabolic profile (e.g. *ALDOB*^high, *FASN*^high), a cluster abundantly expressing drug metabolism genes (e.g. *CYP2C9/19*^high, *ABCC2*^high), as well as a cluster of proliferating hepatocytes (e.g. *MKI67*⁺, *CDK1*⁺). Most markers of fetal and mature cholangiocytes (e.g. *KRT7, MUC5B, FAM178B*) were not expressed (Fig. 2b, Supplementary Fig. 1c).

These analyses strengthened the observation that hepatocyte organoids exclusively emerge upon outgrowth from fetal liver tissue in the specific culture conditions used.

We then performed bulk mRNA sequencing on all collected samples from the temporal FH organoid outgrowth experiment from tissue. Principle component analysis (PCA) revealed significant transcriptomic changes occurring upon plating the liver cells, as well as temporal changes during the following time points (Fig. 1e). We next interrogated the broad transcriptomic changes during organoid outgrowth (Fig. 1f). This revealed 7 distinct hierarchical gene clusters, each displaying unique temporal dynamics (Fig. 1g, h, Supplementary Fig. 2a). Cluster 1 comprised genes whose expression steadily increased with culturing time. GO-term analysis revealed many genes related to translation and respiratory electron transport chain, perhaps reflecting the energy requirements and steady growth of the organoids. Genes in cluster 2 displayed a rapid increase in expression during the early time points, but their expression gradually decreased as their growth advanced into organoids that were ready to be passaged (day 7), yet this response was re-activated one day post passaging (ps+1d). Many of the genes in cluster 2 related to gene expression and DNA replication, suggestive to reflect a dynamic proliferative response. Clusters 3 and 4 comprised genes that were typically associated with markers of non-hepatocyte cells, including hematopoietic cells (*KLF1*)[27], Kupffer cells (*MAFB*)[7], stellate cells (*CYGB*)[8], liver sinusoidal endothelial cells (*CLEC4G*)[7], and fetal cholangiocytes (*FAM178B*)[10], while the mature cholangiocyte marker *KRT7* was not detected. Expression of these genes rapidly (cluster 4) or more gradually (cluster 3) faded with time, since these non-hepatocyte cells are not retained in epithelial organoid culture. Cluster 5 was characterized by a single rapid surge in expression of genes related to cytokine signaling, likely originating from initially-present immune cells. Expression profiles of genes in cluster 6 largely comprised genes involved in cytosolic transport and in the unfolded protein response. Cluster 7 comprised genes mainly related to lipid metabolic processes, including cholesterol and triglyceride biosynthesis, as well as genes related to the complement cascade. Expression profiles of cluster 7 genes were dynamic, with a rapid decrease during the early time points of organoid growth, yet a gradual gain in expression until the organoids were ready to be passaged (day 7), while expression decreased again one day post passaging (ps+1d).

### An inverse transcriptomic relationship between proliferation and lipid metabolism underlies hepatocyte organoid growth

The opposite gene expression trends in tissue clusters 2 and 7 suggested that an inverse relationship between metabolic programs (cluster 7) and proliferation (cluster 2) associates with organoid growth (Fig. 1f, g). Since this tissue dataset did not capture a pure hepatocyte-only response, we designed an experiment to map the early transcriptomic responses upon organoid regrowth from single FHs obtained through single cell dissociation of FH organoid lines (2 donors) (Fig. 2c). We observed widespread transcriptomic changes, peaking at the first 24 h post seeding, with >3000 differentially-expressed genes (DEGs) (Fig. 2d, Supplementary Fig. 3a, b). GO-term enrichment analysis on DEGs identified at 24 h post seeding revealed significant changes in cell cycle and DNA replication genes (upregulated), while genes related to metabolism, including lipid-related processes and cholesterol biosynthesis, were downregulated (Fig. 2e). These observations mirrored the early organoid outgrowth responses from tissue. Visualization of the expression trends of genes contained in tissue clusters 2 and 7 in the current single FH dataset further underscored their similarity, also at later time points (Fig. 2f, Supplementary Fig. 3c). We inspected trends of selected single genes, related to these early reciprocal transcriptomic changes (Fig. 2g, h, Supplementary Fig. 3d). In addition to the characteristic surge in expression of important cell cycle genes (e.g. *CCNE1, CDK1*) and DNA replication

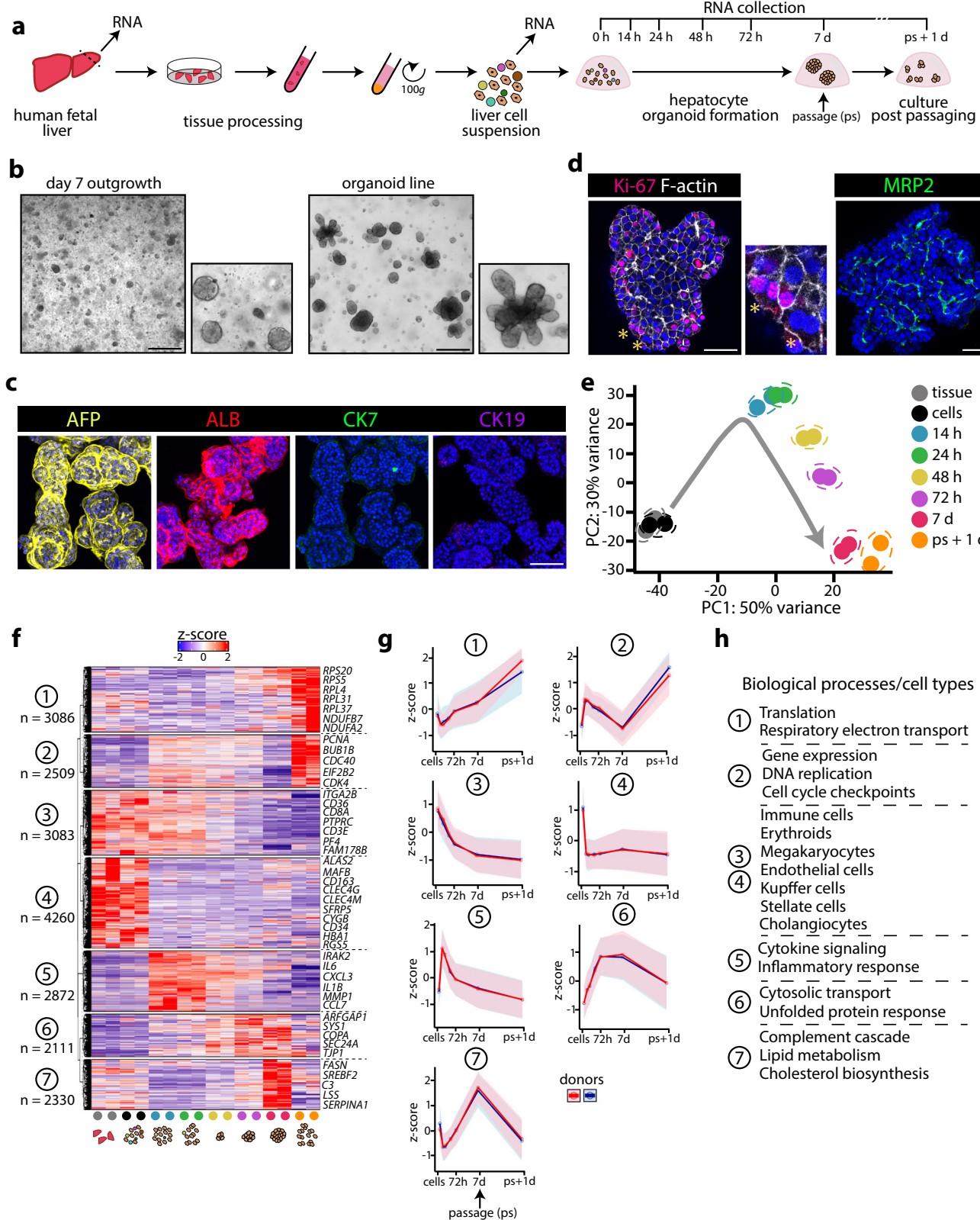

genes (e.g. *PCNA*, *MCM4*), we observed induction of Wnt target genes, (*LGR5, AXIN2, MYC*), highlighting the importance of active Wnt signaling during organoid growth (Fig. 2g). An inverse trend for genes involved in various metabolic processes was observed. Genes involved in de novo lipogenesis (e.g. *SREBF1, FASN*) and cholesterol biosynthesis (e.g. *SREBF2, LSS*) were rapidly repressed upon organoid outgrowth.

We also noted repressed expression of genes related to lipid export and digestion (e.g. *APOB, PNPLA3*) (Fig. 2h), suggesting complex alterations at multiple levels of lipid homeostasis.

To assess putative phenotypic consequences, we assessed the presence of lipid droplets during organoid growth from single cells. Dissociated single FHs displayed minimal lipid droplets, yet many

**Fig. 1 | Temporal transcriptomic characterization of human fetal hepatocyte organoid growth from tissue. a** Experimental strategy to temporally address the transcriptomic changes associated with FH organoid growth from tissue. **b** Representative brightfield images of FH organoids outgrowing from tissue at day 7 post seeding (left) and an established organoid line. Scale bar = 300 μm. **c** Representative images of immunofluorescence staining for AFP, ALB, CK7 and CK19 in FH organoids. Scale bar = 150 μm. **d** Representative image of immunofluorescence staining for Ki-67 and F-actin (left) and MRP2 (right) in FH organoids. Asterisks indicate binucleated hepatocytes. Scale bar = 50 μm. **e** PCA plot visualizing the temporal transcriptomic changes underlying FH organoid growth from tissue across *n* = 2 donors. **f** Heatmap displaying the temporal expression patterns of genes significantly differently expressed at least at one time point versus 0 h (|log2FC| > 0.5, p-adj < 0.05) based on responses of *n* = 2 donors. The expression patterns are visualized as row *Z*-scores. The *n* indicates the number of genes belonging to each cluster. **g** Temporal *Z*-score expression of the genes identified in the different fetal tissue gene clusters across *n* = 2 donors. Mean ± SD is plotted, clusters 1–7: *n* = 3086, 2509, 3083, 4260, 2872, 2111, and 2330 genes, respectively. **h** Biological processes/cell types associated with the different temporal gene clusters, based on GO-term enrichment analysis and manual inspection. **b**–**d** Representative of characterization of *n* = 4 expanding FH organoid cultures. Source data are provided as a Source data file.

cells had accumulated lipids 24 h post seeding. These cells were hypertrophic, yet had not duplicated (Fig. 2i, j, Supplementary Fig. 4a, b). At 48 h, most cells had completed the first round of division and lipid phenotypes rapidly disappeared. From day 3 onwards, the majority of cells continued to form lipid-free multicellular organoids (Fig. 2j, k, Supplementary Fig. 4a, b). These observations mirror transient lipid phenotypes and the extensive metabolic-proliferative transcriptomic rewiring observed upon partial hepatectomy in rodents[28–33]. The origin and relevance of the transient steatosis remains unclear, but a lipid-related epigenetic control on cell cycle genes has been postulated[34].

## Maturation of human fetal hepatocyte organoids based on growth-informed signals

We wondered whether altering growth-associated signals could be exploited to achieve further maturation of FH organoids. We considered Wnt-promoting signals as the most crucial mitogenic factor in the hepatocyte expansion medium promoting proliferation, given the earlier observed Wnt target gene response during organoid outgrowth (Fig. 2g). We assessed the effect of withdrawal of Wnt signals (RSPO1-conditioned medium and the GSK3β inhibitor CHIR-99021 (CHIR)). Removal of RSPO1 did not cause noticeable differences, while CHIR removal led to rapid organoid death (Supplementary Fig. 5a). We then evaluated additional factors that could be needed to induce viable maturation. We focused on HNF4α and CREB, given their central roles in controlling hepatic metabolism processes[35–37], and based on their predicted involvement in organoid growth (Supplementary Fig. 3e). In addition to CHIR removal, we included forskolin (FSK) (increasing cAMP levels) and dexamethasone (DEX) (reported to induce HNF4α in hepatocytes[38]) (Fig. 3a). When we switched FH organoids to this medium (termed maturation medium), organoid morphology rapidly changed into structures with a thick wall (Fig. 3b). Of note, while FSK promotes growth of cholangiocyte organoids[21], FSK addition to the expansion medium of hepatocyte organoids slows down their growth and induces some traits of maturity[39] (Supplementary Fig. 5b). These distinct responses may find their origin in the hepatic cell type-dependent roles of cAMP-PKA signaling[40,41]. In maturation medium, organoids seized to proliferate, yet remained viable for at least 3 weeks. Cells adopted the typical polygonal shape of mature hepatocytes and became visibly larger (Fig. 3c). By quantifying different morphological features of hepatocyte maturity, we found that cell area and nucleus-to-cytoplasm ratio increased, and binucleated cells were more abundant (Fig. 3d–f). Evaluation of the ALB-AFP expression ratio further suggested acquisition of maturity, with a notable reduction in *AFP* mRNA expression (Fig. 3g). On protein level, typical ALB granules more prominently appeared, while AFP was absent in most cells (Fig. 3h). Evaluation of mRNA expression of some functional hepatocyte markers revealed increased expression of multiple *CYPs* as well as the bile acid transporter *SLC10A1* (NTCP) in maturation medium (Fig. 3i). Taken together, further maturation of FH organoids can be achieved by transcriptome-informed interference with growth-related signals.

## Identification of factors influencing human fetal hepatocyte organoid growth

The culture medium used to grow FH organoids has been designed to replace essential non-hepatocyte derived factors, but may be further optimized. Our transcriptomic tissue dataset allowed identifying putative non-hepatocyte derived factors during the early time points of organoid outgrowth when, in addition to the hepatocytes, various other liver cells are still present. Gene cluster 5 was characterized by a transient inflammatory-like response, marked by various interleukins (Figs. 1f, 4a, Supplementary Fig. 2a). Amongst these, IL6 is a well-studied mitogen in mouse liver regeneration[42,43]. Several growth factors also displayed similar mRNA induction patterns. To predict if such factors were hepatocyte-autonomous or non-hepatocyte derived, we compared gene expression trends across our two transcriptomic datasets (Fig. 4b, Supplementary Fig. 6a). The surge in interleukin expression was prominent in the tissue dataset, while expression was low or near-absent in the single FH dataset, arguing for their non-hepatocyte origin. Instead, expression trends of e.g. EGF family ligands were similar in both datasets, suggesting these to be autonomously produced by hepatocytes (Fig. 4b, Supplementary Fig. 6a). We performed cytokine/growth factor challenges to functionally address their importance in organoid regrowth dynamics from single FHs (Fig. 4c). FHs appeared very responsive to all tested factors (Fig. 4d, Supplementary Fig. 6b), and measurement of the diameter of the outgrowing organoids confirmed these observations (Fig. 4e). Supplementation of NRG1 most significantly boosted FH organoid outgrowth, followed by IL6 and IL11 addition. Surprisingly, IL1β, associated with various hepatic diseases[44], likewise boosted FH organoid growth.

## Initial growth of human adult hepatocyte organoids mimics the fetal response yet stalls early

We next focused on organoid outgrowth mechanisms of human adult hepatocytes, which are more difficult to grow as organoids[23]. We applied a similar experimental approach (using the same expansion medium as used for FH organoids) to evaluate their transcriptomic responses utilizing commercial primary human hepatocyte (PHH) sources from *n* = 2 donors (Fig. 5a). Small organoids grew out over the course of 7 days (Fig. 5b), but these structures (ALB+) seized to grow (Ki-67−) and failed to regrow after passaging (Fig. 5c). We then assessed the transcriptomic responses upon PHH organoid growth. PCA revealed major temporal changes during the first 7 days (Fig. 5d, Supplementary Fig. 7a, b). Of note, both PHH donors displayed largely similar transcriptomic trends (and growth arrest) despite their age differences (0.3 vs. 28 years), suggesting that hepatocyte growth differences between development and adulthood are already largely determined rapidly after birth. To assess whether PHH organoid growth associated with the same inverse proliferation-lipid metabolism transcriptomic relationship as observed in the FH organoid datasets, we evaluated expression of genes belonging to fetal tissue clusters 2 and 7. Notably, we observed different dynamics. The early adult responses resembled the fetal response, i.e. an early surge in proliferative signals concomitant with repression of metabolic signals.

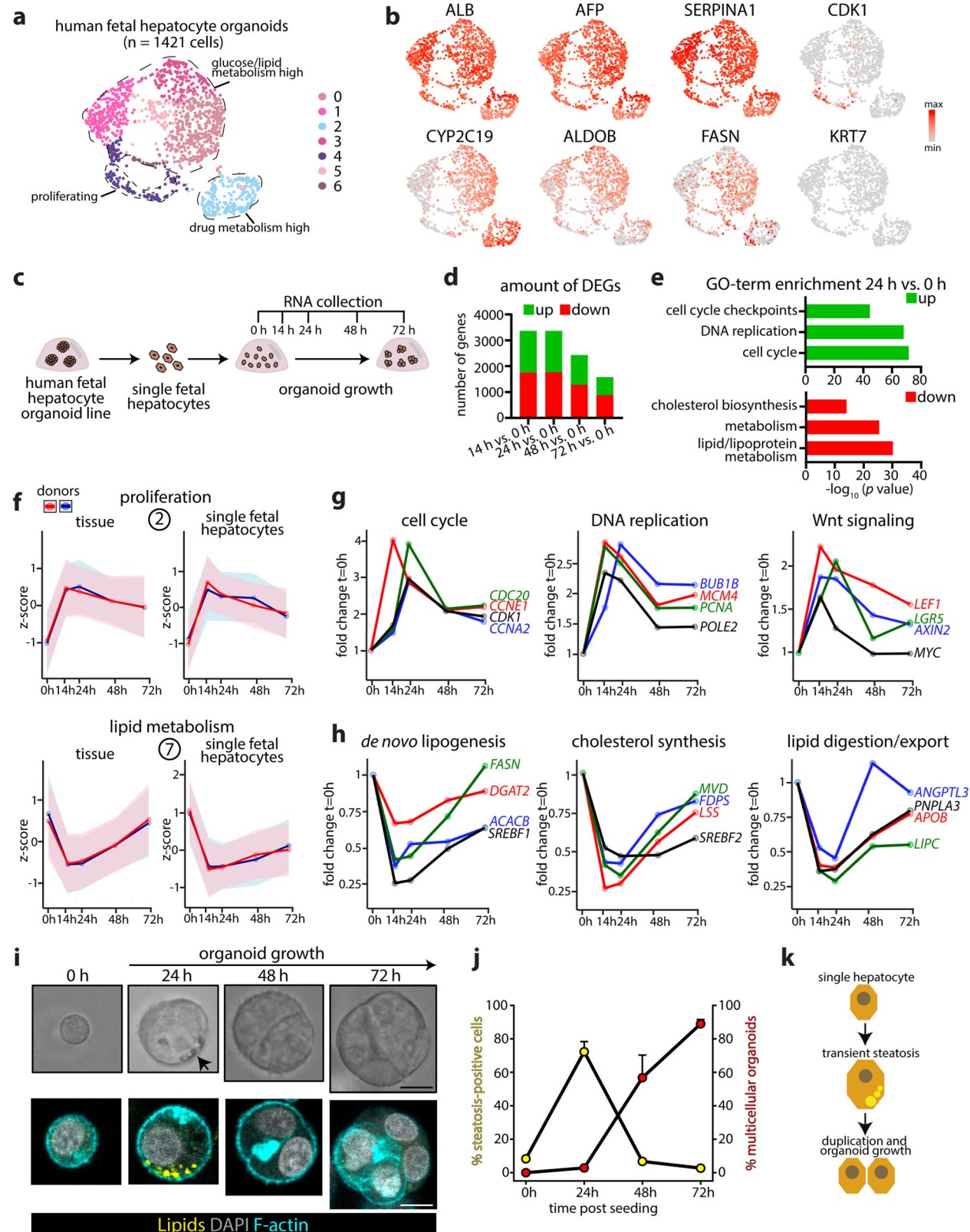

Yet, the re-initiation of these inverse programs early after organoid passaging (ps+1d) was lost and, instead, transcriptomic responses flattened out (Fig. 5e). Visualization of the temporal transcriptomic trends of various proliferation and lipid metabolism genes further corroborated this observation (Fig. 5f), and differed from the fetal dynamics (compare with Supplementary Fig. 3d). Various lipid metabolism-associated genes displayed different temporal dynamics, including a gradual increase in expression of fatty acid metabolism-related genes (e.g. *SCD, FADS1, FADS2*) and cholesterol-related genes (e.g. *SQLE, HMGCR*) (Fig. 5f). *CDKN1A* (p21), a hallmark senescence marker[45], was gradually induced during culture, suggesting its involvement in the growth arrest. The absence of an inverse proliferation-

**Fig. 2 | Rewiring of lipid metabolic programs upon human fetal hepatocyte organoid growth. a** Single-cell profiling of human FH organoids. **b** UMAP plots of the indicated markers. **c** Experimental strategy to temporally address the transcriptomic changes associated with human organoid growth from single FHs. **d** Cumulative plots of the amount of differentially expressed genes per time point post seeding, all versus 0 h (|log2FC| > 0.5, p-adj < 0.05) based on responses of $n = 2$ donors. **e** GO-term enrichment analysis on the upregulated and downregulated DEGs identified at 24 h versus 0 h based on responses of $n = 2$ donors. **f** Temporal $Z$-score expression of the genes identified in fetal tissue clusters 2 and 7 upon organoid growth from single FHs (single cells) across $n = 2$ donors. The organoid growth responses from fetal liver tissue are plotted for comparison. Mean ± SD is plotted, clusters 2 and 7: $n = 2509$ and 2330 genes, respectively. **g, h** Temporal mRNA expression profiles of selected genes involved in the cell cycle, DNA replication, and Wnt signaling (**g**) and selected classes of lipid metabolism genes (**h**). Mean fold change expression of $n = 2$ donors relative to 0 h is shown. **i** Representative brightfield images (top) and lipid staining overlaid with Phalloidin (bottom) of organoid growth from single FHs over time. Representative of $n = 2$ outgrowth experiments. Scale bar = 15 μm. **j** Quantification of the amount of steatosis positive cells during organoid growth (yellow) and the appearance of multicellular organoids (red, defined as $n ≥ 2$ cells/organoid) at different time points post seeding of single FHs. Mean ± SD is plotted, $n = 3$ fields analyzed per time point with a total of $n = 137$ cells (0 h), $n = 135$ cells (24 h), $n = 289$ cells (48 h), and $n = 258$ cells (72 h). **k** Schematic illustrating the transient steatosis occurring upon FH organoid growth from single fetal hepatocytes. Source data are provided as a Source data file.

lipid metabolism response post organoid passaging coincided with their stalled growth.

## Improved human adult hepatocyte organoid growth through rebalancing the cellular metabolic state

We sought to improve PHH organoid expansion. We first asked whether we could exploit the identified growth stimulating factors identified for FH organoids (Figs. 4d, e, 6a). Out of all factors tested, only IL6 supplementation notably boosted PHH outgrowth, resulting in increased organoid diameters (Fig. 6b, c). Yet, these organoids gradually displayed spontaneous accumulation of lipids (Fig. 6d, Supplementary Fig. 8a), and almost no cell divisions were detected upon live imaging (Fig. 6e, Supplementary Movie 1). We asked whether "tweaking" the cellular metabolic state could reverse this apparent senescence. We focused on FXR given its central role in liver metabolism. Strikingly, PHHs supplemented with IL6+FXRa robustly grew out, and accordingly organoid diameters increased (Fig. 6a–c). FXRa supplementation alone influenced organoid outgrowth to a much more limited extent (Fig. 6c). Only under IL6+FXRa supplementation, PHH organoids could regrow after passaging, appearing healthy and without signs of lipid accumulation (Fig. 6b). Accordingly, we detected multiple cell divisions upon live imaging (Fig. 6f, Supplementary Fig. 8b, c, Supplementary Movie 1). Prompted by the observation that the senescence marker p21 was gradually induced in culture (Fig. 5f), we additionally included the BMP antagonist Noggin in this PHH expansion medium for the first few days after passaging, which was beneficial for organoid regrowth. Organoids from young PHHs from 2 donors (0.3 and 1.7 years) cultured under these conditions could be expanded for at least 4 months (Fig. 6g, Supplementary Fig. 8d).

We characterized the PHH organoids under IL6+FXRa supplementation. Some organoids were of dense appearance, while most were thicker-walled with larger lumens (Fig. 7a, Supplementary Movie 2). Individual cells displayed typical polygonal morphology (Fig. 7a, b), and Ki-67 positivity was observed in multiple cells (Fig. 7c). Hepatocyte marker expression was apparent by staining for ALB, while the immature marker AFP was not expressed (Fig. 7d), the latter contrasting the FH organoids (Fig. 1c). Expression of the bile duct markers CK19 and CK7 was not detected (Fig. 7d). The organoids were further broadly positive for A1AT, HNF4α, and CYP3A4, and the tight junction marker ZO1 demarked their complex polarity (Fig. 7e, Supplementary Fig. 8e).

We then surveyed the cellular identity of an established PHH organoid culture from one donor by single-cell RNA sequencing, recovering 3413 cells (Fig. 7f, Supplementary Fig. 9a, b). These analyses revealed broad expression of typical hepatocyte markers (e.g. *ALB, TTR, RBP4, TF, HP*) (Fig. 7g, Supplementary Fig. 9c). Mature hepatocytes (*AHSG*^high^, *CES1*^high^) were abundant and expressed cytochrome P450s (e.g. *CYP2D6, CYP2E1*). Within these, a coagulation-high population was apparent marked by abundant expression of *F9, F10, F12*, and *SERPINC1*. A smaller progenitor-like population (*PROM1*^+^) was also identified. Proliferating hepatocytes were marked by e.g. *MKI67* and *CENPF* positivity. Expression of the mature cholangiocyte markers

*KRT7, AQP1* and *CFTR* was mostly absent or confined to some cells in the progenitor-like population (Fig. 7g, Supplementary Fig. 9c). Abundant expression of IL6 target genes (e.g. *APP, SAA2, SERPINA3*) and the absence of expression of *CYP7A1* and *CYP17A1* (genes repressed by FXR signaling) further corroborated active IL6+FXRa signaling (Supplementary Fig. 9c). Finally, direct comparison of the single-cell profiles of the PHH organoids with those of the FH organoids (Fig. 2a) revealed their distinct cellular identities (Fig. 7h). We evaluated hepatocyte marker genes reported to change in expression pattern from development to adulthood[10]. This revealed selective expression of typical fetal (e.g. *AFP, GPC3, MT1G*) or adult (e.g. *APCS, NNMT, CYP2E1*) hepatocyte markers across the respective organoid datasets, corroborating preservation of age identity in culture (Fig. 7i, Supplementary Fig. 9d). Altogether, these comparative analyses and functional experiments identified different growth requirements of fetal and adult human hepatocytes in organoid culture (Fig. 7j).

## Discussion

Here, we have constructed time-resolved transcriptomic maps of organoid growth initiated from fetal and adult human hepatocytes. Using identical culture conditions, this allowed to directly highlight key growth differences and commonalities, serving as a proxy to estimate hepatocyte growth mechanisms during development and adulthood/regeneration. We found that hepatocyte organoid growth associates with extensive reciprocal transcriptomic rewiring of metabolic and proliferative programs. Exploiting these insights, we devised culture conditions to mature FH organoids by acting on proliferative and metabolic hubs. While FH organoids can be long-term expanded[23,25], organoids initiated from PHHs in the same culture condition displayed rapid growth arrest and altered proliferation-lipid metabolism transcriptomic dynamics. In addition, we noted substantial discrepancies between the responses of fetal and adult human hepatocytes to exposure to growth factors and cytokines. While FHs versatilely responded to multiple interleukins and EGF family ligands (i.e. enhanced growth), PHHs only notably responded to IL6. The need for this proinflammatory cytokine matches the observation for the need of another proinflammatory cytokine, TNFα, for mouse hepatocyte organoid growth[24]. IL6 is a widely studied mitogen in the context of partial hepatectomy[46,47], and IL6 endows robust mouse hepatocyte expansion in 2D culture[43]. Our data thus suggest that human hepatocytes in adulthood display reduced pan-sensitivity to growth stimuli and become more specific in their requirements for growth.

IL6 supplementation could boost initial PHH organoid growth, but lipid accumulation and growth arrest followed. Simultaneous activation of FXR prevented the induction of a steatotic state, and translated to robust organoid growth. Lipid metabolism and proliferation are well-known intricately linked processes[48] and imbalanced metabolism in liver diseases is associated with impaired cell division[49,50]. FXR is a master hepatic metabolic regulator, controlling both lipid and bile acid processes[51]. FXR has been less explored in the context of liver regeneration. Whole-body *Fxr* knock-out mice

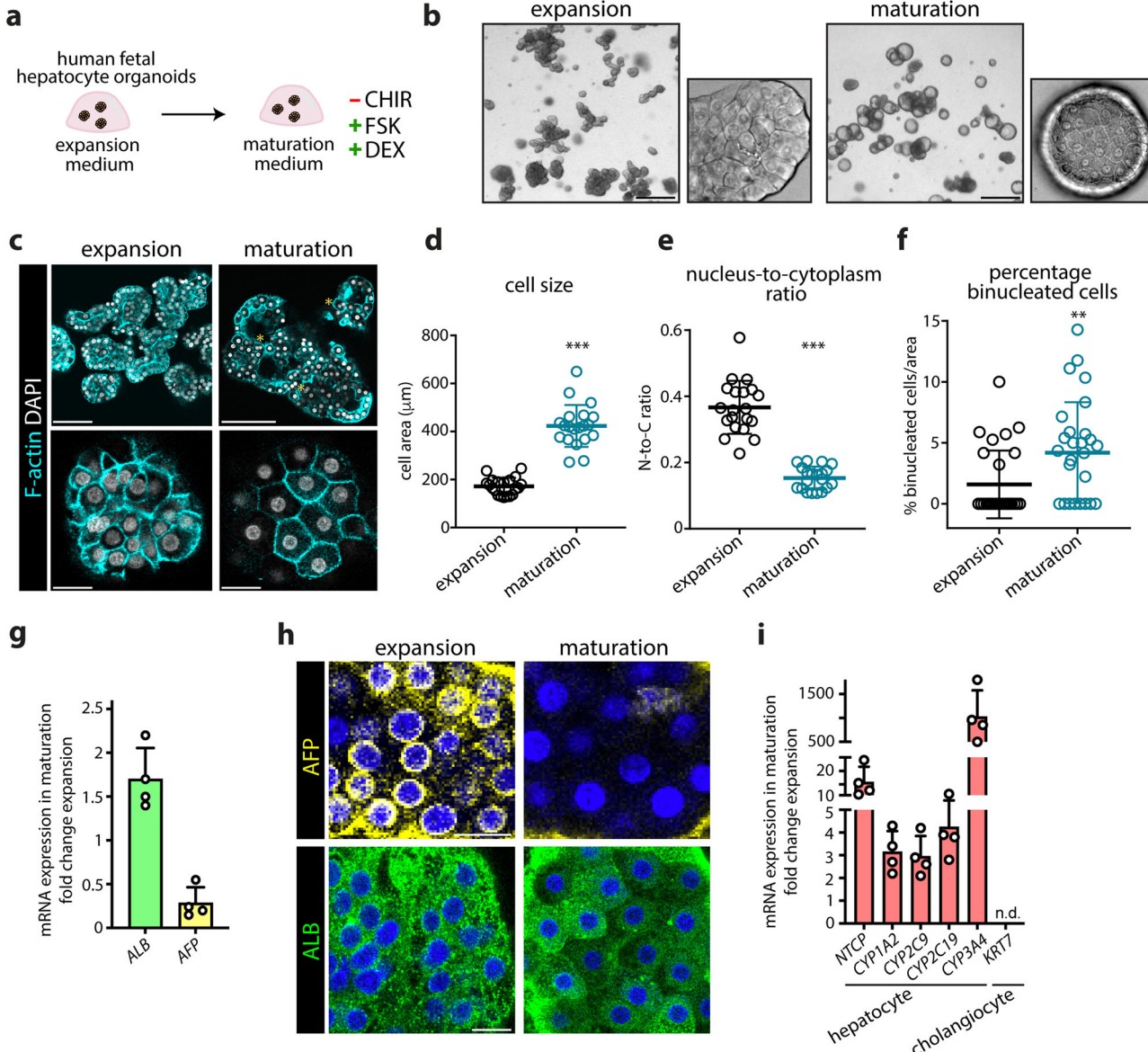

**Fig. 3 | Modulating culture conditions matures human fetal hepatocyte organoids. a** Experimental strategy to mature FH organoids. Altered factors in the maturation medium are indicated. **b, c** Representative brightfield images (**b**) and phalloidin staining (**c**) of FH organoids in expansion medium and maturation medium (14 days post switch). Asterisks indicate binucleated hepatocytes. Scale bar = 400 µm (**b**), 150 µm (low mag) and 30 µm (high mag) (**c**). **d–f** Quantification of cellular features of FH organoids in expansion and maturation medium based on phalloidin staining, including cell area (**d**), nucleus-to-cytoplasm ratio (**e**), and the percentage of binucleated cells (**f**). Mean ± SD is plotted with n = 20 cells per condition (**d, e**) and n = 28 quantified areas per condition (**f**). Maturation versus

expansion: ***p < 0.0001, two-tailed unpaired t-test (**d, e**), **p = 0.0083, Mann–Whitney U test (**f**). **g, h** mRNA expression of AFP and ALB (**g**) and immunofluorescence staining for AFP and ALB (**h**) in FH organoids in maturation medium relative to expansion medium. Mean ± SD is plotted, n = 4 matured organoid cultures (**g**). Scale bar = 25 µm (**h**). **i** mRNA expression of different functional hepatocyte markers in FH organoids in maturation relative to expansion medium. Mean ± SD is plotted, n = 4 matured organoid cultures. n.d. not detected. **b, c, h**, Representative of characterization of n = 2 expanding and matured FH organoid cultures. Source data are provided as a Source data file.

displayed delayed liver regeneration after partial hepatectomy[52], linked to both intestinal and hepatic actions[53]. Interestingly, FXR activation improved age-related proliferation defects in regenerating mouse livers[54]. FXR is currently investigated as a therapeutic target in metabolic dysfunction-associated steatotic liver disease[55]. In agreement, we previously noted beneficial effects of FXR activation in organoid models of steatosis[56]. Our current data imply that FXR activation may not only be beneficial for resolution of diseased metabolic phenotypes, but may also concomitantly improve liver regeneration in conjunction with IL6 treatment.

Recent studies have made substantial progress in the expansion of human adult hepatocytes in vitro[57–60]. In 2D culture, significant

expansion of PHHs can be achieved using strategies based on their conversion into a bipotent state, evident by both morphological analysis and marker expression, and these cells can be readily re-differentiated into hepatocyte-like cells and can repopulate host livers[57–61]. We here defined a culture condition that allows improved 3D organoid expansion from PHHs, which retain multiple hepatocyte features, including typical polygonal morphology and broad expression of key markers. As we employed young PHH donors, future benchmarking across multiple donors with more diverse demographics will help inform on their broad suitability for application in drug discovery and regenerative purposes. In conclusion, we show how transcriptomic knowledge can be exploited to design growth and

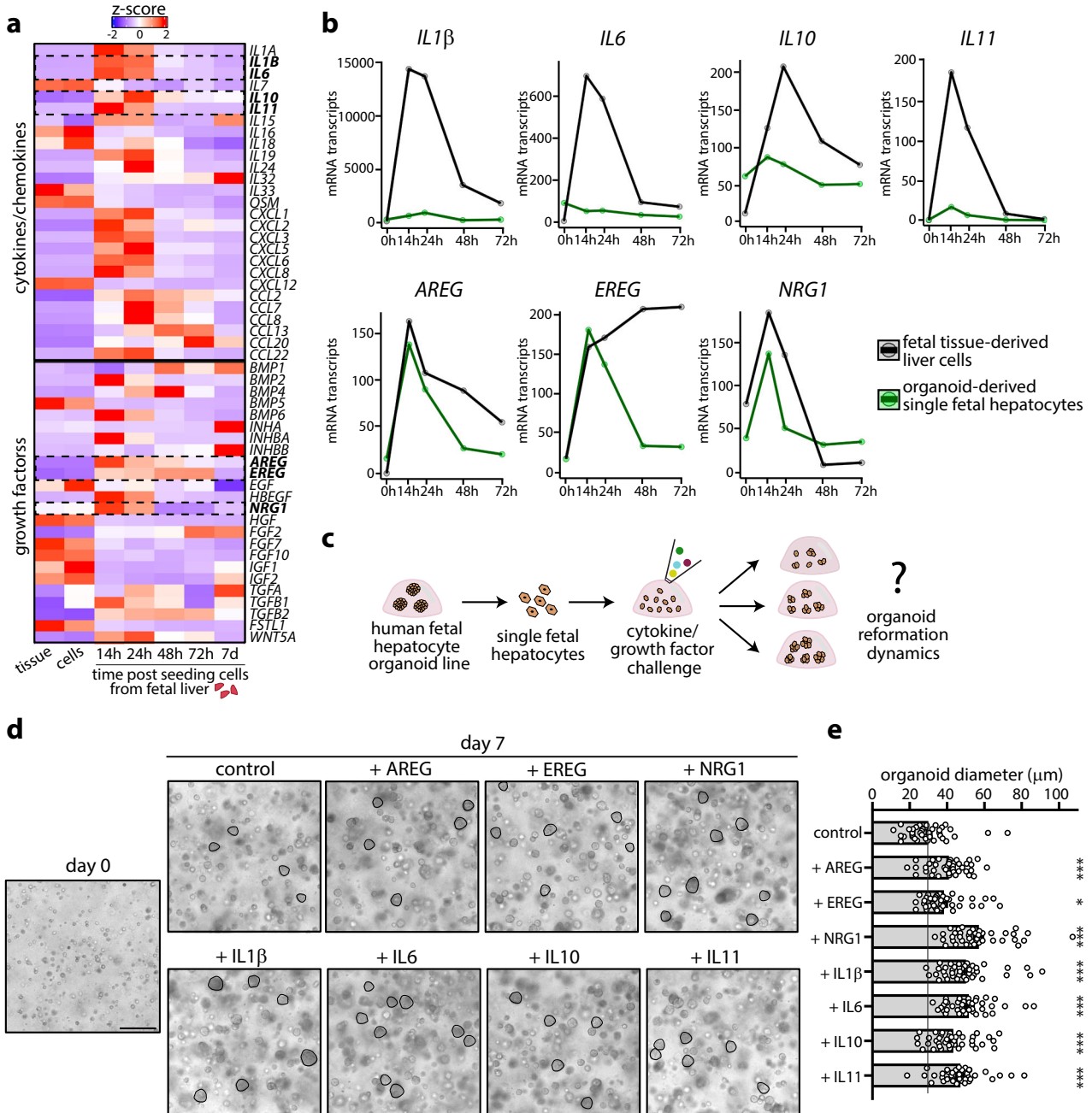

**Fig. 4 | Identification of autocrine and paracrine signals that boost human fetal hepatocyte organoid growth. a** Heatmap displaying gene expression patterns of cytokines, chemokines, and growth factors upon FH organoid growth from tissue. The mean expression of $n = 2$ donors is visualized as row Z-scores. **b** Comparison between the temporal mRNA expression profiles of selected cytokines and growth factors upon hepatocyte organoid growth from tissue (black) or single FHs (green). The mean expression of $n = 2$ donors per dataset is plotted (normalized transcripts). **c** Experimental strategy to evaluate the effect of cytokines and growth factors on human FH organoid growth. **d** Representative brightfield images of organoid-derived single FHs at day 0 and the outgrowing organoids challenged with indicated cytokines and growth factors at day 7. Representative of $n = 2$ challenge experiments. Scale bar = 100 μm. **e** Quantification of the organoid diameter 7 days post organoid outgrowth from single FHs under the different challenges ($n = 40$ organoids per condition). Mean ± SD is plotted. EREG: *$p = 0.0143$; AREG: ***$p = 0.0005$; NRG1, IL1β, IL6, IL10, IL11: ***$p < 0.0001$, all versus control, one-way ANOVA with Dunnett's post hoc test. Source data are provided as a Source data file.

maturation protocols and to improve culture conditions for hard to grow cell types.

## Methods

### Human fetal hepatocyte organoid establishment from tissue

Human fetal liver tissues were derived from healthy abortion material from anonymous donors, under informed consent and ethical approval (Commission of Medical Ethics, Leiden University Medical Center, Leiden). FH organoid lines were established from tissue (FH donors 1–4, GW10, GW12, GW15, GW12, respectively) and cultured, largely as described previously[25,39], with some modifications as follows. Liver cell suspensions were obtained by a short collagenase IV (Sigma–Aldrich) incubation with subsequent mincing of the tissue. The resulting cell suspension was retrieved after a $500\,g$ spin to ensure cellular diversity. The cells were washed twice with cold Advanced DMEM/F12 (Thermo Fisher, 12634010) supplemented with 1x

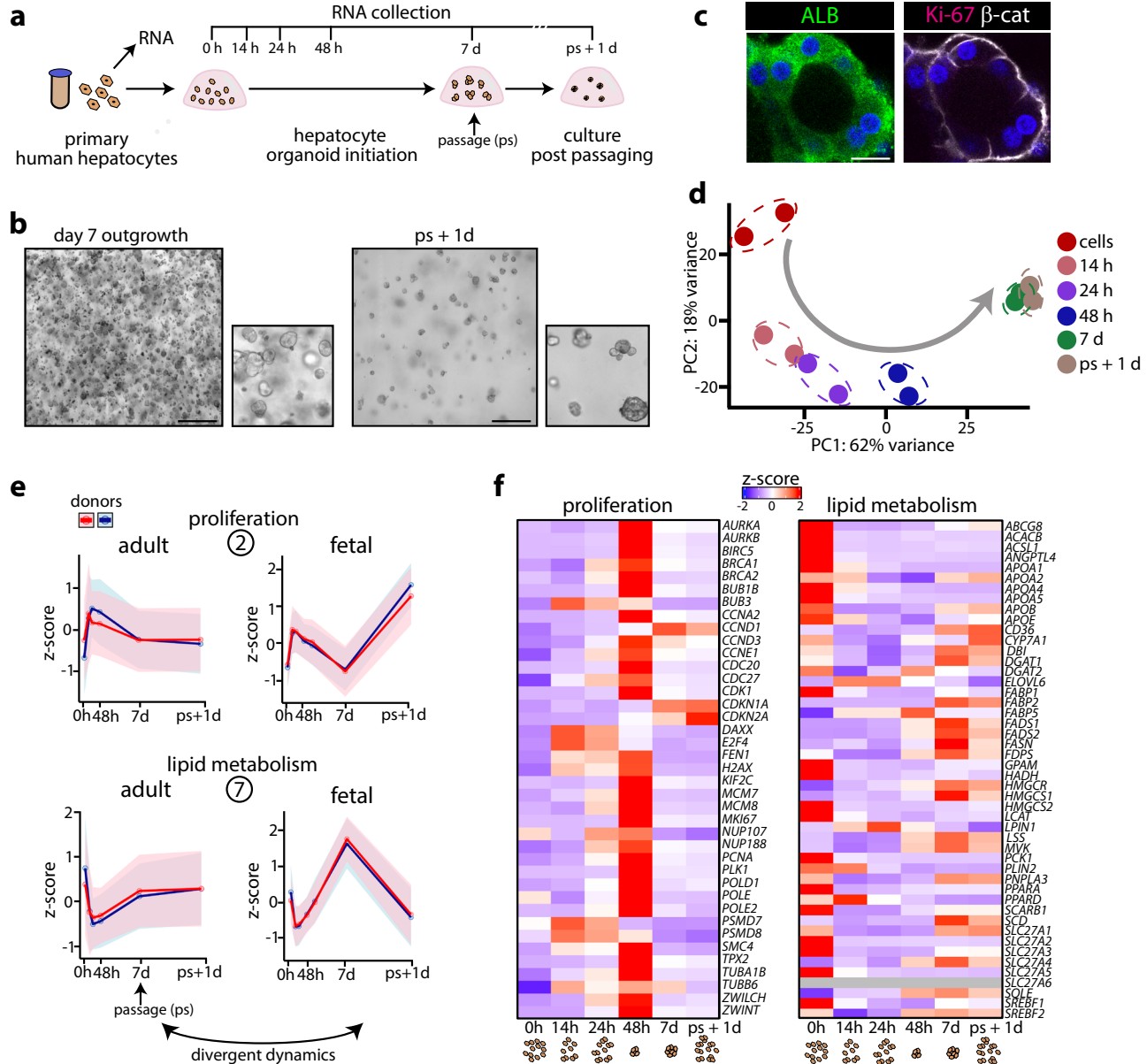

**Fig. 5 | Temporal transcriptomic characterization of organoid growth from primary human hepatocytes. a** Experimental strategy to temporally address the transcriptomic changes associated with organoid growth from PHHs.
**b** Representative brightfield images of outgrowing PHH organoids at day 7 post seeding. Scale bar = 300 μm. **c** Representative images of immunofluorescence staining for ALB, Ki-67, and β-catenin protein in PHH organoids. Scale bar = 25 μm. **d** PCA plot visualizing the temporal transcriptomic changes underlying PHH organoid growth across $n = 2$ donors. **e** Temporal Z-score expression of the genes

identified in the fetal tissue clusters 2 and 7 during PHH organoid growth. The FH organoid growth responses from fetal liver tissue (see Fig. 1g) are plotted for comparison. Mean ± SD is plotted, clusters 2 and 7: $n = 2509$ and 2330 genes, respectively. **f** Heatmaps displaying gene expression patterns of proliferation-related genes and lipid metabolism-related genes during PHH organoid growth. The mean expression trends of $n = 2$ donors are visualized as row Z-scores.
**b, c** Representative of characterization of $n = 2$ outgrowing PHH organoid cultures. Source data are provided as a Source data file.

GlutaMAX (Thermo Fisher, 35050-061), 10 mM HEPES (Thermo Fisher, 15630-056) and 100 U/ml penicillin/streptomycin solution (Thermo Fisher, 15140-122), from now on referred to as AdvDMEM/F12+++. Cells were then plated into domes of basement membrane extract (BME) (Cultrex, 3533-005-02) in a 2:1 ratio of BME:AdvDMEM/F12+++ (33 μl/ drop) with 3 drops per well of a 12-well plate. Cells were cultured in FH expansion medium, which consisted of AdvDMEM/F12+++ supplemented with 15% RSPO1-conditioned medium (in-house production), 1x B-27 Supplement Minus Vitamin A (Thermo Fisher, 12587010), 2.5 mM nicotinamide (Sigma−Aldrich, N0636), 1.25 mM N-acetyl-L-cysteine (Sigma−Aldrich, A9165), 50 ng/ml EGF (Peprotech, AF-100-15), 50 ng/ml HGF (Peprotech, 100−39), 20 ng/ml TGFα (Peprotech, 100-

16 A), 10 nM gastrin (Sigma−Aldrich, G9145), 3 μM CHIR-99021 (Sigma−Aldrich, SML1046), 5 μM A 83-01 (Tocris, 2939), and 50 μg/ml primocin (InvivoGen, ant-pm1). This medium was supplemented with 10 μM Y-27632 (AbMole, M1817) to minimize anoikis during the first few days of culture. Organoids were cultured at 37 °C and 5% $CO_2$ and passaged using gentle pipetting 1:2 every 10−14 days. Cultures were regularly tested for mycoplasma and tested negative without exception.

### Organoid growth from single human fetal hepatocytes
Organoids from established FH organoid lines were collected from the BME domes using cold AdvDMEM/F12 +++, centrifuged at 100 g,

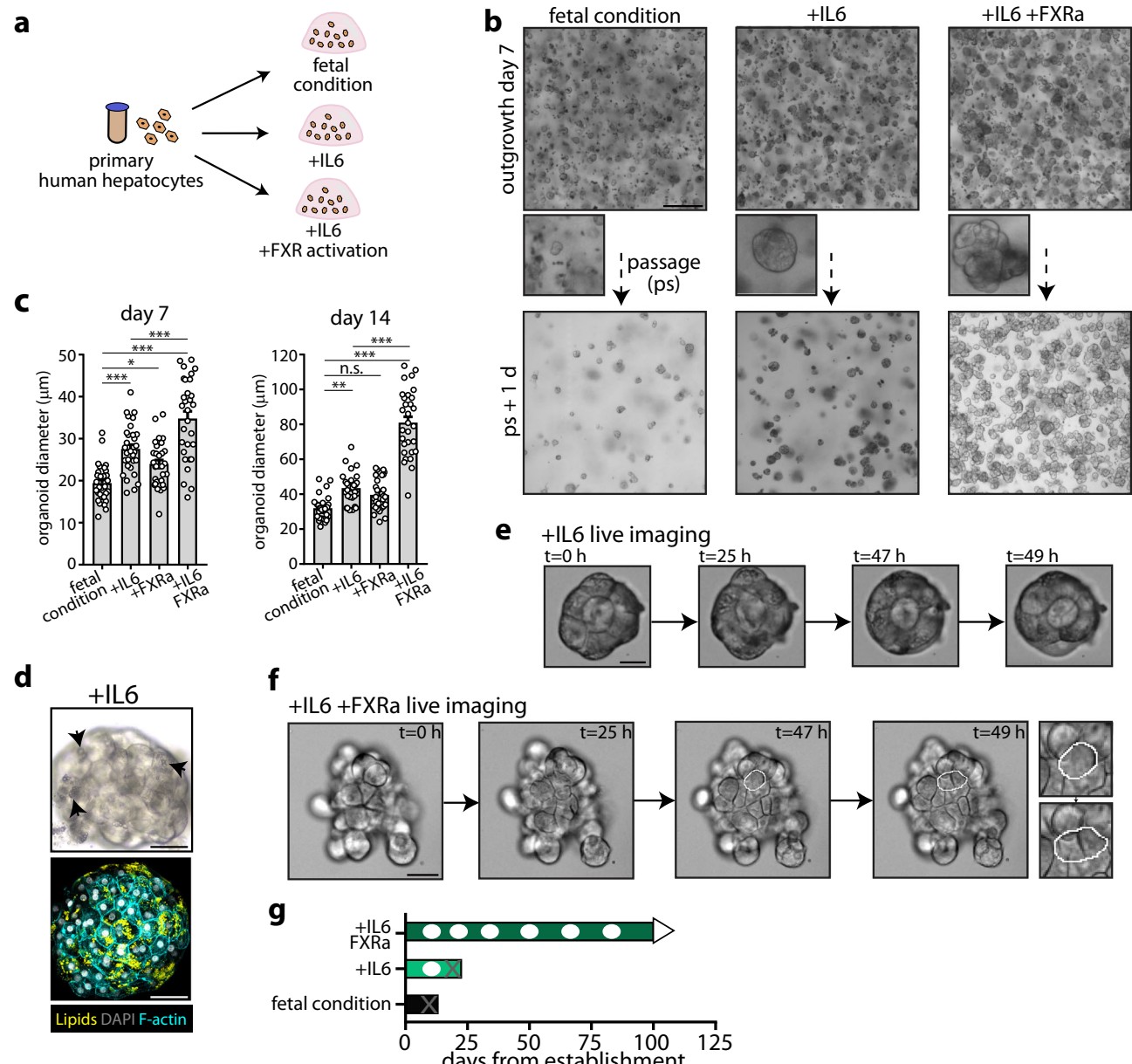

**Fig. 6 | Synergy between IL6 and FXR activation boosts primary human hepatocyte organoid growth. a** Experimental strategy to test different organoid growth conditions for PHHs. **b** Representative brightfield images of outgrowing PHH organoids under fetal culture conditions, and upon supplementation of IL6 and IL6+FXRa at day 7 post seeding. Scale bar = 300 μm. **c** Quantification of the diameter of outgrowing PHH organoids at day 7 and 14 post seeding under the different conditions. Mean ± SD is plotted, $n = 30$ organoids per condition. FXRa versus fetal condition, day 7: *$p = 0.0441$; IL6 versus fetal condition and IL6+FXRa versus fetal condition, day 7: ***$p < 0.0001$; IL6+FXRa vs IL6, day 7: ***$p = 0.0003$; IL6 versus fetal condition, day 14: **$p = 0.0013$, IL6+FXRa versus fetal condition and IL6+FXRa versus IL6, day 14: ***$p < 0.0001$, one-way ANOVA with Tukey post hoc

test. n.s. not significant. **d** Representative brightfield image and lipid staining overlaid with phalloidin of an IL6-cultured organoid. Arrows point at extensive lipid accumulation. Scale bar = 20 μm. **e, f** Representative time-lapse images visualizing the divergent growth potencies of an IL6-cultured organoid (**e**) and an IL6+FXRa cultured organoid (**f**). White outlines highlight cell division. Scale bar = 20 μm (**e, f**). **g** Growth characteristics of PHH organoids when cultured under the different conditions. White dots indicate passaging. Grey crosses indicate terminated cultures (failure to regrow after passaging). **d–f** Representative of characterization of $n = 2$ expanding PHH organoid cultures. Source data are provided as a Source data file.

and made into single cells (single FHs) using a 5 min. Accutase (Thermo Fisher, 00-4555-56) treatment on the resulting cell pellet with intermittent gentle pipetting. After two washes with cold AdvDMEM/F12 +++, the single FHs were seeded into domes of BME and cultured in FH expansion medium (supplemented with 10 μM Y-27632) to allow organoid growth from single FHs. The outgrowing cells/organoids were temporally harvested for downstream analyses, including transcriptomic characterization of organoid outgrowth

from single fetal hepatocytes and the associated phenotypic characterization of lipid profiles.

## Maturation of human fetal hepatocyte organoids
Expanding FH organoids were collected from the BME domes using cold AdvDMEM/F12+++ and made into smaller organoid fragments by gentle pipetting. The organoid fragments were washed twice with cold AdvDMEM/F12+++, replated in BME domes, and were subsequently

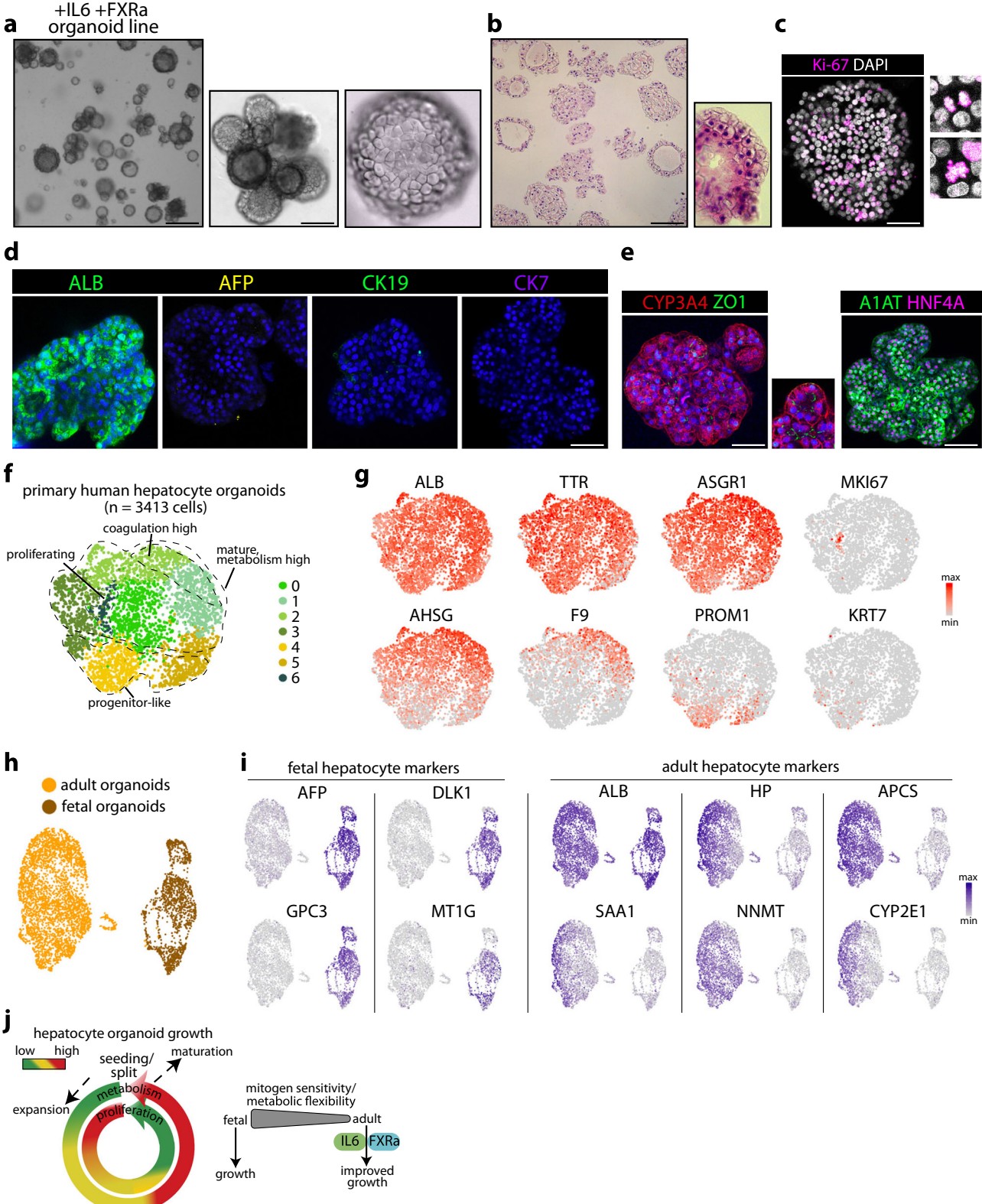

**Fig. 7 | Characterization of primary human hepatocyte organoid cultures.**
**a** Representative low- and high-magnification brightfield images of expanding PHH organoids (passage 4) in the IL6+FXRa condition. Scale bar = 400 μm (low mag) and 100 μm (high mag). **b** Representative image of H&E staining of PHH organoids. Scale bar = 75 μm. **c** Representative immunofluorescence staining for Ki-67 of PHH organoids. Scale bar = 100 μm. **d, e** Representative immunofluorescence staining of PHH organoids for (**d**) ALB, AFP, CK19, and CK7 and (**e**) A1AT and HNF4A, CYP3A4, and ZO1. Scale bar = 100 μm (**d**, **e**). **f** Single-cell profiling of PHH organoids. **g** UMAP plots of the indicated markers. **h** Single-cell profiles of the FH and PHH organoids (originating from a single library, with the two cultures distinguished based on genotype demultiplexing). **i** UMAP plots of the indicated fetal and adult hepatocyte markers in the combined single cell datasets. **j** Schematic illustrating the dynamics underlying human hepatocyte growth and differences between fetal and adult. **a–e** Representative of characterization of *n* = 2 expanding PHH organoid cultures.

cultured in FH maturation medium, which was composed of FH expansion medium but without inclusion of CHIR-99021 and with addition of 10 µM forskolin (R&D Systems, 1099) and 0.2 µM dexamethasone (Sigma–Aldrich, D4902). Matured organoids were harvested for analysis (qPCR, immunofluorescence) 14 days post maturation. Quantification of cell area, nucleus-to-cytoplasm ratio, and percentage of binucleated cells were performed ImageJ (Fiji) software (v2.14.0). In brief, the cell area was measured by outlining individual cell areas denoted by F-actin (Phalloidin) staining. In parallel, the area of each matched nucleus (DAPI+) was measured (considering only mono-nucleated cells). These two measures were used to calculate the nucleus-to-cytoplasm ratio. Different fields ranging between a size of 10–45 cells were used to count the percentage of binucleated cells over total cells measured in each field. Multiple organoids ($n > 10$ per condition) were used for quantitative analysis.

## Cytokine and growth factor challenges

Human FH organoids were made into single FHs as described above. Single FHs were plated into domes of BME and grown in FH expansion medium (control), or FH expansion medium supplemented with either 20 ng/ml IL10 (Peprotech, 200-10), 20 ng/ml IL11 (Peprotech, 200-11), 20 ng/ml IL1β (Peprotech, 200-01B), 100 ng/ml IL6 (Peprotech, 200-06), 100 ng/ml NRG1 (Peprotech, 100-03), 250 ng/ml AREG (Peprotech, 100-55B), or 250 ng/ml EREG (Peprotech, 100-04). Organoid growth was temporally evaluated and brightfield pictures were acquired for further analysis. Organoid diameters 7 days post outgrowth were measured using ImageJ (Fiji) software (v2.14.0).

## Primary human hepatocyte organoid establishment

PHHs were obtained from commercial sources (Lonza, HUCPG and HUCPI; Sigma-Aldrich, MTOXH10002), PHH donors 1-3, 0.3, 28, 1.7 years, respectively). Upon thawing of the cells according to manufacturer's instructions, cells were seeded into domes of BME in a 2:1 ratio of BME:AdvDMEM/F12+++ (33 µl/drop), with 2 drops per well of a 24-well plate. The FH expansion medium as described above for FH organoids was used for transcriptomic experiments to evaluate PHH organoid outgrowth. To optimize PHH organoid growth, different conditions were tested, including supplementation of the FH expansion medium with combinations of 100 ng/ml IL6, 20 ng/ml IL11, 20 ng/ml IL1β, or 100 ng/ml NRG1, and 10 µM cilofexor (GS-9674, FXRa) (MedChemExpress, HY-109083). The optimized PHH expansion medium consisted of AdvDMEM/F12 +++ supplemented with 15% RSPO1-conditioned medium, 1x B-27 Supplement Minus Vitamin A, 2.5 mM nicotinamide, 1.25 mM N-acetyl-L-cysteine, 50 ng/ml EGF, 50 ng/ml HGF, 20 ng/ml TGFα, 10 nM gastrin, 3 µM CHIR-99021, 5 µM A 83-01, 50 µg/ml primocin, 100 ng/ml IL6, and 10 µM FXRa. This medium was supplemented with 10 µM Y-27632 to minimize anoikis during the first few days of culture. From passage 1, 1.5% Noggin-Fc conditioned medium (UPE, N002) was added to this medium for the first 5–8 days after passaging. Upon culture establishment, fibroblast outgrowth was avoided by initial replating of the cells and contaminating outgrowing large and thin-walled duct-like cystic organoids were meticulously manually removed from the culture. Organoids were cultured at 37 °C and 5% $CO_2$ and passaged using gentle pipetting 1:2 every 10–14 days for the first month, and every 2–3 weeks thereafter. Cultures were regularly tested for mycoplasma and tested negative without exception.

## Immunofluorescence staining

Immunofluorescence staining of organoids was performed essentially as described previously[25]. Briefly, organoids were fixed with 4% PFA for 2 hours at RT and washed twice with PBS. Organoids were blocked and permeabilized with 5% BSA-PBS including 0.2% Triton-X for 1 hour at RT. Organoids were washed once with PBS and incubated with primary antibody diluted in 2.5% BSA-PBS O/N at 4 °C. Primary antibodies

included anti-AFP (Thermo Fisher, PA5-16658, 1:250); anti-ALB (Bethyl, A80-229A, 1:500); anti-Ki-67 (Thermo Fisher, 14-5698-82, 1:1000); anti-CK7 (Thermo Fisher, MA5-11986, 1:400); anti-CK19 (Cell Signaling Technology, 13092 S, 1:500); anti-MRP2 (Abcam, ab3373, 1:500); anti-BCAT (Santa Cruz, sc-7199, 1:500); anti-A1AT (Bethyl, A80-122A, 1:250); anti-HNF4A (Abcam, ab201460, 1:250); anti-CYP3A4 (Thermo Fisher, MA5-17064, 1:250); anti-ZO1 (Thermo Fisher, PA5-19090, 1:250). Organoids were washed with PBS three times and subsequently incubated with the appropriate secondary Alexa Fluor antibodies diluted in 2.5% BSA-PBS for 3 hours at RT. After one wash with PBS, organoids were incubated with 1 µg/ml DAPI (Thermo Fisher, 62248), and cell membranes were optionally stained with Phalloidin-Atto 647N (Sigma–Aldrich, 65906, 1:1000), both in PBS for 20 min at RT. After two washes with PBS, organoids were transferred to a 96-well Sensoplate and imaged on a Leica Sp8 microscope.

## Lipid staining

Lipid staining was performed on fixed organoids by performing a 20 min incubation at RT with 1 µg/ml Nile Red (Sigma–Aldrich, 72485), while membranes were stained with Phalloidin-Atto 647N (1:1000), and nuclei were counterstained with DAPI (1 µg/ml). After two washes with PBS, organoids were imaged as described above. Quantification of steatosis positivity was evaluated based on the clear presence of lipid droplets within the cells/organoids and quantification of mono-cellular/multicellular organoids was performed based on the Phalloidin-nuclei outline of cells/organoids, both performed using ImageJ (Fiji) software (v2.14.0).

## RNA extraction and qPCR analysis

RNA was extracted using Trizol (Thermo Fisher, 15596-018). RNA was reverse transcribed into cDNA using the Superscript IV kit (Thermo Fisher, 18091050). qPCR analysis was performed using iQ SYBR Green Supermix (Thermo Fisher, 1708887) and the specific primers listed in Supplementary Data 1.

## Bulk RNA sequencing

For temporal transcriptomic profiling of organoid growth responses, the following cultures were used: FH donor 1 (GW10) and FH donor 2 (GW12) for organoid growth from fetal liver tissue, FH donor 1 (GW10) and FH donor 3 (GW15) for organoid growth from organoid-derived single fetal hepatocytes, PHH donor 1 (0.3 years) and PHH donor 2 (28 years) for organoid growth from PHHs. RNA integrity and concentrations were measured using the Agilent RNA 6000 Nano kit (Agilent, 5067-1511) and Qubit RNA HS Assay Kit (Thermo Fisher, Q32852), respectively. All samples were of RIN >8. Library preparation and sequencing were performed by USEQ (Utrecht Sequencing Facility). RNA libraries were prepared with the TruSeq Stranded messenger RNA polyA kit and paired-end (2 × 75 base pairs) sequenced on an Illumina NextSeq 500. Reads were mapped to the human GRCh37 genome assembly. Lowly expressed genes were filtered out (<10 transcript counts across all samples) and transcript normalization was performed using DESeq2[62] (v1.36.0) in RStudio environment. The plotPCA function from DESeq2 was used to generate PCA plots. Differential gene expression analysis was performed using DESeq2, employing paired analysis to take into account donor information. DEGs were considered when $|log2FC| > 0.5$ and p-adj < 0.05. Gene Ontology (GO)-term enrichment analyses and predicted transcription factor involvement were performed using Enrichr[63]. Data visualization was performed using the packages ggplot2, ComplexHeatmap and EnhancedVolcano in RStudio, or manually plotted using GraphPad Prism (v9.4.1).

## Single-cell RNA sequencing

FH organoids from FH donor 4 (GW12) (5 months in culture) and PHH organoids from PHH donor 3 (1.7 years) (3 months in culture) were

processed in parallel to construct a single library. Organoids were made into single cells using Accutase, as described above. The dissociated cells were washed twice with HBSS without calcium and magnesium. To enhance viability, we employed the Dead Cell Removal kit with MS columns (Miltenyi, 130-090-101) according to the manufacturer's instructions. Live cells were manually counted using Trypan blue and resuspended in PBS containing 0.04% BSA, mixing both samples into one solution (>80% viability). The library was prepared according to the 10x Genomics manufacturer's construction and sequenced on an Illumina NovaSeq 6000, both performed by the Single Cell Genomics facility of the Princess Máxima Center. Mapping and UMI counting was performed using Cell Ranger software (v7.1.0). Genotype deconvolution (i.e. to distinguish the fetal and adult cells) was performed using souporcell[64] (v2.0). Further bioinformatic analyses were performed using Seurat[65] (v4.3.0) in RStudio environment. A SeuratObject was created and the genotype metadata was added using the function AddMetaData. Cells with <1000 detected genes, >6500 genes, or >25% mitochondrial content were removed. Data was normalized using the NormalizeData function using a scale.factor of 10,000. Variable features were identified using the function FindVariableFeatures using the standard method vst (features = 2000). Data was scaled using the ScaleData function and dimensionality of the dataset was determined using the function ElbowPlot. Clusters were determined with the functions FindNeighbors (dims = 1:15) and FindClusters (resolution = 0.5). Unassigned and doublet cells were removed based on the genotype metadata (n = 4834 cells). This joint dataset was used to compare fetal and adult hepatocyte marker expression. Each dataset was also independently analyzed by subsetting the object based on the genotype metadata and using the same cluster resolution (0.5). This resulted in n = 1421 cells and n = 3413 cells for the FH and PHH organoid datasets, respectively. Data visualization was performed using UMAP plots and violin plots using Seurat. The function FindAllMarkers was used to find markers for the different clusters in both datasets and the top 10 markers per cluster were visualized in heatmaps.

### Statistics and reproducibility

No statistical method was used to predetermine sample size, no data were excluded from the analyses, the experiments were not randomized, and the investigators were not blinded to allocation during experiments and outcome assessment. For the bulk transcriptomic profiling of hepatocyte organoid growth, n = 2 donors were used across (i) fetal liver tissue (FH donor 1, FH donor 2), (ii) single FHs (FH donor 1, FH donor 3), and (iii) PHHs (PHH donor 1, PHH donor 2). Evaluation of factors influencing FH organoid growth and maturation conditions were evaluated across n = 2 donors (FH donor 1, FH donor 4). Testing of optimized growth conditions for PHH organoids was performed across n = 2 donors (PHH donor 1, PHH donor 3). Single-cell RNA sequencing of both fetal and adult hepatocyte organoids was performed using each a single donor (FH donor 4, PHH donor 3). Statistical analyses were performed using GraphPad Prism or DESeq2. Sample sizes (n), statistical tests, p values and considered statistical significance are indicated in the figure legends.

### Reporting summary

Further information on research design is available in the Nature Portfolio Reporting Summary linked to this article.

## Data availability

Read-level bulk and single-cell RNA sequencing data generated in this study have been deposited in the GEO database under accession code GSE264262. The associated raw data are protected and are not available due to data privacy laws. Source data are provided with this paper.

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

## Acknowledgements

We thank Stieneke van den Brink for RSPO1-conditioned medium production, Harry Begthel for histology, and Dr. Ype de Jong for sharing resources. D.H. is supported by a VENI grant from the Dutch Research Council (NWO, VI.Veni.212.134).

## Author contributions

D.H. and B.A. conceived the study. D.H., B.A. and H.C. supervised the study. D.H. and B.A. designed and conducted experiments, analyzed and interpreted data, and performed data visualization. T.M. aided with single-cell RNA sequencing experiments. I.Z. provided technical

support. S.C.d.S.L. shared resources. D.H., B.A. and H.C. wrote the manuscript.

## Competing interests

Near the end of this study, H.C. became head of Pharma, Research and Early Development (pRED) of F. Hoffmann-La Roche Ltd, Basel, Switzerland. H.C. holds several patents on organoid technology. Their application numbers, followed by their publication numbers (if applicable), are as follows: PCT/NL2008/050543, WO2009/022907; PCT/NL2010/000017, WO2010/090513; PCT/IB2011/002167, WO2012/014076; PCT/IB2012/052950, WO2012/168930; PCT/EP2015/060815, WO2015/173425; PCT/EP2015/077990, WO2016/083613; PCT/EP2015/077988, WO2016/083612; PCT/EP2017/054797, WO2017/149025; PCT/EP2017/065101, WO2017/220586; PCT/EP2018/086716, n/a; and GB1819224.5, n/a. D.H., B.A. and H.C. are inventors on a filed patent related to hepatocyte organoids (PCT/NL2022/050641, n/a). The other authors declare no competing interests.
