## [Peer Review File · Nature Communications]

Mapping of mitogen and metabolic sensitivity in organoids defines requirements for human hepatocyte growthREVIEWER COMMENTS

Reviewer #1 (Remarks to the Author):

The finding that removal of CHIR; and that addition of Forskolin and Dex to cultures promotes the maturation of fetal hepatocytes are important findings. The findings that IL-6, NGR1 and IL11 promoted the growth of fetal organoid cultures are also of interest. In contrast, IL1B inhibited but IL-6 promoted the growth of adult hepatocytes. However, there are several concerns that need to be addressed to improve the paper and strengthen the conclusions of the paper.

1. Bulk RNA sequencing was performed on the cultures, but (as was admitted by the authors) the strength of the findings would be greatly increased if scRNA-Seq was used to evaluate the cultures. As the data shows, there is significant cellular heterogeneity in the cultures, especially at the early stages. The authors highlighted the high level of tissue complexity'. Therefore, to better understand hepatocyte regeneration and niche cells, scRN-Seq data is strongly preferred. Although when discussing the data for cytokine expression, the text indicates that single cell and tissue data was evaluated, but it is unclear what data was compared.

1. There is a critical need to improve the clarity of the text. The abstract and introduction is very difficult to read. It is important to state more clearly what is known, what is not known and then indicate what was found. Abbreviations (BME, CHIR) are used throughout the text without an indication of what they stand for. For example, I had to read the methods to infer that CHIR was CHIR-99021, which is an inhibitor of GSK3b. The terms 'expansion medium', 'HEP' medium 'fetal culture condition' and 'standard' medium are used. Are they the same thing? In the methods, in 'Human fetal hepatocyte organoid regrowth from single cells', what is 'using a ca' and 'AdvDMEM/F12+++'? What's p1 and p2 cells referred, if its passage, why use different passages? how about 'line'?

2. The authors need to clarify the origin and more carefully characterize the fetal-liver derived organoids. Why did the culture conditions only generate hepatocytes but not other epithelial cells such as cholangiocytes? Using a very similar method, the authors previously could generate bi-potent cells. Was the origin of starting cells only hepatocytes, or did they contain Igr5+ cells and bile duct cells?

3. In a previous paper (<https://doi.org/10.1016/j.cell.2014.11.050>), 'FSK addition upregulated LGR5 and the ductal marker KRT19, while ALB and CYP3A4 decreased.' This is contradictory to the effect of FSK using in 'mature medium' in this manuscript.

4. The 'reciprocal interaction between proliferation and lipid metabolism' should be further investigated. With the tissue data, the lowest proliferation was on day 7 and the highest level of lipid metabolism is also on day7. However, with the single cell formation experiments, only day 3 results are shown. So extended experiments are needed to confirm the claim. Also, de novo lipogenesis and cholesterol biosynthesis are repressed on day 1, while digestion and export genes were repressed. This leads to the

question of where the excess lipids come from. Hence, other key lipid transporter genes should be examined, which include: SLC27A1, SLC27A2, SLC27A3, SLC27A4, SLC27A5, SLC27A6, CD36, FABP1, FABP2, FABP5, DBI, and ACSL1.

6. How did authors distinguish where those niche factors came from, is it because of stress induced or just from dying immune cells?

Reviewer #2 (Remarks to the Author):

The authors report on studies of human organoids to analyze the growth of fetal and adult hepatocytes through temporal transcriptomic and phenotypic approaches. Following the establishment of organoids from the human fetal liver, they show that cells claimed to be derived from “the niche” release inflammatory cytokines such as IL6, while fetal hepatocytes independently express growth factors like NRG1. Enhancing the presence of these factors facilitates the formation of fetal organoids. Throughout the outgrowth process, individual fetal hepatocytes initiate reciprocal transcriptional programs characterized by increased proliferation and suppressed lipid metabolism.

Examining adult hepatocytes, they report that cells initially follow the fetal trajectory but then exhibit imbalanced dynamics between proliferation and lipid metabolism, leading to steatosis and growth cessation. Supplementation of IL6 and simultaneous activation of the metabolic regulator FXR prevent steatosis and stimulate the proliferation of adult hepatocytes, resulting in the establishment of expanding human adult hepatocyte organoid lines.

The paper has two main results:

- 1) Specific mitogen requirements and metabolic distinctions that dictate the proliferation of hepatocytes
- 2) The ability to consistently expand hepatocytes from adult origin into organoids, based on the activation of FXR

While the authors are somewhat concise in describing their data, it should be noted that most of the manuscript is very descriptive. There are many statements that don't lead to a conclusion, and leave the reader with guessing what the relevance is. For example: “Thus, fetal hepatocytes enter a self-regulated “transient steatosis” stage prior to the first duplication event. A proliferation-metabolism transcriptional network appeared to be temporally rewired to allow organoid regrowth from a single fetal hepatocyte”. What is meant with “ a network that appeared to be temporally rewired”?

Here an example of a list of genes that are expressed without much of an impact:

“We noted transient expression of various interleukins, including IL1B, IL6, IL10 and IL11. Amongst these, IL6 is a well-studied mitogen in the context of hepatocyte regeneration the mouse liver^{21,22}. Some growth factors also displayed a similar mRNA induction, including the EGF-related ligands amphiregulin”

It is not clear what is meant by the "niche" in the context of the liver. Whether there are stem cell niches

in the normal liver, sources of self-renewing signals and even the existence of hepatocyte stem cells are issues that have not been clarified by earlier work, nor does this paper shed light on.

I recommend publication of a shortened version of this work, emphasizing the two main results.

Reviewer #1 (Remarks to the Author):

The finding that removal of CHIR; and that addition of Forskolin and Dex to cultures promotes the maturation of fetal hepatocytes are important findings. The findings that IL-6, NGR1 and IL11 promoted the growth of fetal organoid cultures are also of interest. In contrast, IL1B inhibited but IL-6 promoted the growth of adult hepatocytes. However, there are several concerns that need to be addressed to improve the paper and strengthen the conclusions of the paper.

A: We thank the Reviewer for his/her constructive comments. We have taken into account all remaining concerns, which helped us to improve our manuscript. Our replies to each point are detailed below.

1. Bulk RNA sequencing was performed on the cultures, but (as was admitted by the authors) the strength of the findings would be greatly increased if scRNA-Seq was used to evaluate the cultures. As the data shows, there is significant cellular heterogeneity in the cultures, especially at the early stages. The authors highlighted the high level of tissue complexity'. Therefore, to better understand hepatocyte regeneration and niche cells, scRN-Seq data is strongly preferred. Although when discussing the data for cytokine expression, the text indicates that single cell and tissue data was evaluated, but it is unclear what data was compared.

A: We agree that single-cell RNA sequencing analysis of the organoid cultures would be of value to better understand the cultures identities.

We have thus now performed single-cell RNA sequencing analysis using the 10X Genomics platform of both the established human fetal hepatocyte organoid cultures, as well as the (now optimized) adult primary human hepatocyte organoid cultures.

The scRNA-seq analyses of the fetal organoids highlight their uniform hepatocyte identity (AFP+, ALB+, SERPINA1+, TTR+, etc.). In contrast, markers of fetal/adult cholangiocytes such as FAM178B, KRT7, and AQP1 are absent, while the progenitor marker KRT19 is sparsely expressed in some cells. Within the dataset, we identify a proliferating hepatocyte cluster, as well as a cluster of cells high in hepatocyte drug metabolic capacity (e.g. CYP2C9, CYP2C19), and a cluster of hepatocytes with abundant expression of genes involved in broad metabolic pathways. These novel data are included in Fig. 2a-c and new Supplementary Figure 1.

For the adult organoid dataset, we present the new data in new Fig. 7e-f and Supplementary Fig. 9. Here, we likewise identify broad and abundant expression of hepatocyte markers across all cells (e.g. ALB, TTR, ASGR1, RBP4). A large cluster of cells displayed typical mature hepatocyte characteristics (AHSG+, CES1+ and high expression of various cytochrome P450s, such as CYP2D6 and CYP2E1). Within this cluster we noted F9+, F12+ cells, which represent cells displaying a high coagulation profile (a specific and mature function of the hepatocytes in the liver). We also note a smaller hepatocyte progenitor-like cluster, which express the main hepatocyte markers, but also the progenitor markers PROM1 and KRT19. Mature cholangiocyte markers, such as KRT7, MUC5B, AQP1, CFTR etc. are either absent or lowly expressed confined to the progenitor-population.

Having established these two single-cell datasets also allowed us to assess fetal versus the adult hepatocyte marker expression across the two organoid cultures (based on markers changing from development to adulthood reported by Wesley et al. Nature Cell Biology, 2022).

This revealed that fetal hepatocyte organoid cultures retain markers characteristic of development (e.g. GPC3, MT1G). Instead, adult hepatocyte markers (e.g. NNMT, APCS, CYP2E1) are abundantly expressed in the adult hepatocyte organoid cultures. These data are included in new Fig. 7g-h and Supplementary Fig. 9d.

We believe these additional characterizations allowed us to significantly strengthen our observations and provide valuable resources to evaluate these hepatocyte organoid cultures.

In human organoid in vitro culture and with the specific culture medium used, as we previously published for the fetal hepatocyte organoids in the initial paper from our group, Hu et al. Cell 2018, we are able to specifically grow out hepatocyte organoids from fetal liver tissue, while other non-hepatocyte cells (such as non-parenchymal liver cells) are not retained/propagated in organoid culture, and quickly vanish in culture (which is accordingly visualized in our bulk RNA-sequencing temporal analyses of markers of e.g. non-parenchymal cells). Tissue-derived organoid cultures are indeed optimized to expand solely epithelial cells. It is for this reason that we chose to use bulk RNA-sequencing for the temporal analysis to primarily understand the hepatocyte-based mechanisms that are put into place by the fetal hepatocyte to grow out as an organoid culture. We thus considered scRNA-seq not very advantageous for this temporal experimental set-up in organoids.

1. There is a critical need to improve the clarity of the text. The abstract and introduction is very difficult to read. It is important to state more clearly what is known, what is not known and then indicate what was found. Abbreviations (BME, CHIR) are used throughout the text without an indication of what they stand for. For example, I had to read the methods to infer that CHIR was CHIR-99021, which is an inhibitor of GSK3b. The terms ‘expansion medium’, ‘HEP’ medium ‘fetal culture condition’ and ‘standard’ medium are used. Are they the same thing? In the methods, in ‘Human fetal hepatocyte organoid regrowth from single cells’, what is ‘using a ca’ and ‘AdvDMEM/F12+++’? What’s p1 and p2 cells referred, if is passage, why use different passages? how about ‘line’?

A: We sincerely apologize for the suboptimal clarity of the text. We have now rigorously amended the text, both in the Results and Methods section, to improve its clarity. We now carefully detail all abbreviations used in the text and what the compounds used biologically act on (e.g. CHIR-99021, forskolin, etc.). We have also amended a consistent and clear nomenclature for the culture medium, cells, compounds, etc. used in the different experiments. In addition, as pointed out by the Reviewer, we have paid particular attention to the abstract and introduction to simplify and mainstream the messages of the manuscript and to better describe the state of the field and what we report in this study. The Introduction section now also elaborates on the different culture conditions previously published to culture bile duct or hepatocyte cells as separate organoid cultures with their own specific culture medium.

2. The authors need to clarify the origin and more carefully characterize the fetal-liver derived organoids. Why did the culture conditions only generate hepatocytes but not other epithelial cells such as cholangiocytes? Using a very similar method, the authors previously could generate bi-potent cells. Was the origin of starting cells only hepatocytes, or did they contain Igr5+ cells and bile duct cells?

A: We previously established two distinct culture conditions to grow epithelial cells from the liver. In earlier work (Huch et al. Cell 2015), a specific expansion medium was described which allows the selective expansion of cholangiocyte/bile duct cells as organoids with bipotent capacity, while in our more recent work (Hu et al. Cell 2018), we designed a growth factor cocktail (with a critical factor being the addition of the GSK3 β inhibitor CHIR-99021) to selectively expand and grow fetal hepatocytes as organoids (and not cholangiocytes). In these studies, the origin of these two different cultures has been experimentally addressed. Specifically, fetal hepatocyte organoids were shown to be derived from AFP+ hepatocytes (Hu et al. Cell, 2018). This was further confirmed through genetic fluorescent labelling of the AFP locus, by deriving clonal AFP::mNEON hepatocyte cultures in Artegiani et al. Nature Cell Biology, 2020). In this study, we derive fetal hepatocyte organoids based on Hu et al. Cell, 2018. We have clarified this in the Introduction.

Regarding further characterization of the fetal hepatocyte organoids, we have now performed scRNA-seq analysis to better characterize this culture system (new Fig. 2a-b and Supplementary Fig. 1). We refer to our answer to the first comment, which describes in detail these results and highlights the hepatocyte identity of the cultures (ALB+, AFP+, TTR+, SERPINA1+, etc.), while cholangiocyte markers such as KRT7 and AQP1 are absent. We note expression of the stem cell marker/Wnt target gene LGR5 in some cells within the dataset, and this expression pattern overlaps with other Wnt target genes (e.g. LEF1), reflecting active Wnt signaling (stimulated by CHIR-99021).

3. In a previous paper (<https://doi.org/10.1016/j.cell.2014.11.050>), 'FSK addition upregulated LGR5 and the ductal marker KRT19, while ALB and CYP3A4 decreased.' This is contradictory to the effect of FSK using in 'mature medium' in this manuscript.

A: In the previous paper mentioned by the Reviewer, our group previously established culture conditions with a specific medium to grow and expand cholangiocyte/bile duct cells as 3D organoids (with its own, very different expansion medium including indeed forskolin (FSK)). In that paper, a maturation medium was designed to mature those cells towards a hepatocyte fate (the medium cocktail for this approach did indeed not contain FSK, and overall is substantially different from the media in the current study). The presence of FSK in the cholangiocyte/bile duct expansion medium results in increased growth of these bile duct cell-derived organoids (Huch et al. Cell, 2015). Instead, in the current study, we culture fetal hepatocyte organoids in hepatocyte-specific expansion medium, as previously established in our group (Hu et al. Cell, 2018). We now find that FSK addition to the maturation medium designed in this study for these fetal hepatocyte organoids instead promoted their maturation. To further showcase this, we have performed novel experiments in which we evaluated the effect of FSK addition to the expansion medium of human fetal hepatocyte organoids and evaluated the suggested markers (new Supplementary Fig. 5b). This showed that LGR5 expression is repressed upon FSK addition, reflecting slowing down of hepatocyte organoid growth, while CYP3A4 expression is induced, showcasing some induction of maturation already by FSK addition alone. ALB expression is unaltered (highly expressed in baseline), and KRT19 expression is unaltered (and which remains lowly expressed).

These new data further clarify the differences between the reported effects of FSK on cholangiocyte organoids by Huch et al. and our reported effects on FSK on hepatocyte organoids. The differential response towards addition of FSK of cholangiocyte/bile duct

organoids vs hepatocyte organoids may be explained by the different actions of cAMP-PKA signaling in these two different epithelial cells in the liver (FSK being a cAMP activator). While cAMP-PKA signaling is known as pro-proliferative factor to promote biliary duct cell growth in vivo (Francis et al. J. Hepatol. 2004), in hepatocytes cAMP-PKA signaling is involved in controlling more mature functions including lipid and drug metabolism (Wahlang et al. Cell Signal 2018). We have now included a discussion point on this aspect in the Results section and included references to these papers.

4. The 'reciprocal interaction between proliferation and lipid metabolism' should be further investigated. With the tissue data, the lowest proliferation was on day 7 and the highest level of lipid metabolism is also on day 7. However, with the single cell formation experiments, only day 3 results are shown. So extended experiments are needed to confirm the claim. Also, de novo lipogenesis and cholesterol biosynthesis are repressed on day 1, while digestion and export genes were repressed. This leads to the question of where the excess lipids come from. Hence, other key lipid transporter genes should be exam, which include: SLC27A1, SLC27A2, SLC27A3, SLC27A4, SLC27A5, SLC27A6, CD36, FABP1, FABP2, FABP5, DBI, and ACSL1.

A: We appreciate the suggested additional markers to analyze, and accordingly evaluated these mentioned markers, which are now included in the heatmaps in Supplementary Fig. 3e-f. These further investigations reveal that most of these markers are likewise temporally repressed during organoid outgrowth, while we note only a modest temporal induction of FABP5 and SLC27A4. We agree with the Reviewer that the different lipid metabolism signals are somewhat complex to interpret, given the repression at both lipid digestion/export and at the same time at synthesis level. Nonetheless, the accumulation of lipids reflects a functional outcome related to the prominent transcriptomic changes in hepatocyte lipid metabolism. The expansion medium to grow human fetal hepatocyte organoids does not contain substantial amounts of fatty acids, and we thus consider it unlikely that an increased import of those small sources of fatty acids is the cause of the observed transient lipid phenotypes. Rather, it most likely results from a disturbed lipid digestion/export, resulting in transient lipid accumulation. We have revised the text in the discussion related to this point.

With regard to the dynamics between the tissue outgrowth and single cell formation experiments, we intended to show that similar transcriptional changes are recapitulated in the growth of organoids from single dissociated cells as we observed when initiating organoid growth starting from tissue, concerning the initial decrease of lipid metabolism genes accompanied by a concomitant increase in expression of proliferation genes during the early timepoints upon organoid growth. This is the reason why we analyzed organoid outgrowth transcriptomes in the single cell experiment until day 3, because this timeframe is sufficient to generate small organoids and we indeed found this inverse lipid metabolism-proliferation trend recapitulated in this experimental setting. As requested by the Reviewer, we now also analyzed later time points, that in line with our observation on tissue, show a further increase of lipid metabolism and associated return of proliferation genes to baseline during later timepoints (Supplementary Fig. 3d). We see indeed the highest expression of lipid metabolism genes and lowest expression of proliferation genes at the moment that organoids are fully grown and ready to be passaged.

6. How did authors distinguish where those niche factors came from, is it because of stress induced or just from dying immune cells?

A: Well taken. Our comparative data on fetal liver tissue-derived hepatocyte organoid growth (Fig. 1) versus hepatocyte organoid regrowth from single fetal hepatocytes (Fig. 2) allowed us to distinguish between cell-autonomous versus non-hepatocyte derived signals which are initially present in the tissue (Fig. 4a-b, Supplementary Fig. 6a). This led us to confirm that signals such as the interleukin surge must be non-hepatocyte derived. It is likely that these factors come from initially-present (dying) immune cells, and not from stress-induced signals occurring in hepatocytes. We have toned down the claims and amended the text to describe these observations, describing these as “non-hepatocyte derived factors”. Of note, surges in interleukins are also observed in vivo upon partial hepatectomy and have been assigned to be derived from Kupffer cells (Taub Nat Rev Mol Cell Biol, 2004). We now cite this paper.

Reviewer #2 (Remarks to the Author):

The authors report on studies of human organoids to analyze the growth of fetal and adult hepatocytes through temporal transcriptomic and phenotypic approaches. Following the establishment of organoids from the human fetal liver, they show that cells claimed to be derived from “the niche” release inflammatory cytokines such as IL6, while fetal hepatocytes independently express growth factors like NRG1. Enhancing the presence of these factors facilitates the formation of fetal organoids. Throughout the outgrowth process, individual fetal hepatocytes initiate reciprocal transcriptional programs characterized by increased proliferation and suppressed lipid metabolism.

Examining adult hepatocytes, they report that cells initially follow the fetal trajectory but then exhibit imbalanced dynamics between proliferation and lipid metabolism, leading to steatosis and growth cessation. Supplementation of IL6 and simultaneous activation of the metabolic regulator FXR prevent steatosis and stimulate the proliferation of adult hepatocytes, resulting in the establishment of expanding human adult hepatocyte organoid lines.

The paper has two main results:

- 1) Specific mitogen requirements and metabolic distinctions that dictate the proliferation of hepatocytes
- 2) The ability to consistently expand hepatocytes from adult origin into organoids, based on the activation of FXR

While the authors are somewhat concise in describing their data, it should be noted that most of the manuscript is very descriptive. There are many statements that don't lead to a conclusion, and leave the reader with guessing what the relevance is. For example: “Thus, fetal hepatocytes enter a self-regulated “transient steatosis” stage prior to the first duplication event. A proliferation-metabolism transcriptional network appeared to be temporally rewired to allow organoid regrowth from a single fetal hepatocyte”. What is meant with “ a network that appeared to be temporally rewired”?

A: Well taken. The scope of our manuscript was to describe and understand mechanisms and requirements of human hepatocyte growth in culture as organoids, which we then used to achieve better growth and maturation of human fetal hepatocyte organoids as well as improved adult primary human hepatocyte organoid growth. Some of our findings are indeed more observational than directly functional. The observation of a transient steatosis during hepatocyte organoid outgrowth intrigued us, given that this is a well-known phenomenon described upon hepatectomy in vivo in mouse liver (e.g. Rudnick et al. Int J Hepatol, 2002), which we found to be recapitulated in in vitro culture. Coinciding with the transient steatosis, we observed an inverse transcriptional relationship between the upregulation of various proliferation genes and the concomitant downregulation of multiple lipid metabolism genes. We agree that calling this a “network” was not ideal. We have now changed this wording to “relationship”.

Taking this comment into account, we have rigorously revised (and shortened where possible) the manuscript to provide more concrete and relevant statements. In our view, also the establishment of the novel maturation condition for the human fetal hepatocyte organoids as

a consequence of the findings on the metabolic and mitogen requirements is an important finding of our paper and we therefore highlighted this aspect as well.

Here an example of a list of genes that are expressed without much of an impact:

"We noted transient expression of various interleukins, including IL1B, IL6, IL10 and IL11. Amongst these, IL6 is a well-studied mitogen in the context of hepatocyte regeneration the mouse liver^{21,22}. Some growth factors also displayed a similar mRNA induction, including the EGF-related ligands amphiregulin"

A: We agree that some of the genes mentioned were not very relevant to the study. We have therefore refined and shortened the text to better highlight the key findings. With regard to IL6, we specifically mention this mitogen as an interesting "hit", given the observed importance of IL6 (in combination with FXR activation) to promote hepatocyte organoid growth (Fig. 4d-e, Fig. 6b-c).

It is not clear what is meant by the "niche" in the context of the liver. Whether there are stem cell niches in the normal liver, sources of self-renewing signals and even the existence of hepatocyte stem cells are issues that have not been clarified by earlier work, nor does this paper shed light on.

A: Well taken. Indeed, the "niche" in general remains a controversial topic within liver biology. We intended with the word niche to refer to non-hepatocyte-derived elements required for tissue maintenance. In this regard, we evaluated our bulk RNA-sequencing data to attempt to understand which signaling and factors are expressed during the organoid outgrowth from fetal tissue (in which initially all liver cell elements were present) (Fig. 1). Through comparing this dataset to the bulk RNA-sequencing data from organoid growth from single fetal hepatocytes (Fig. 2), we could identify non-hepatocyte factors and those being likely hepatocyte-autonomous derived, which we initially termed deriving from the "niche" (Fig. 4a-b, Supplementary Fig. 6a). We have now amended this description into "non-hepatocyte-derived factors" and toned down (and shortened) the claims regarding niche contributions.

I recommend publication of a shortened version of this work, emphasizing the two main results.

A: We thank the Reviewer for his/her evaluation of our manuscript. We have amended and shortened the text, taking into account the comments from this Reviewer as well as the comments of Reviewer 1. Please note that we did include additional experiments, characterizations, and textual clarifications, as requested by Reviewer 1. These include novel single-cell RNA sequencing data to describe the identity of both human fetal and adult hepatocyte organoid cultures (new Fig. 2a-b, Fig. 7e-h and Supplementary Fig. 1 and 9), evaluation of the effect of forskolin on hepatocyte organoids, and extended analyses on the inverse lipid metabolism-proliferation trend.

REVIEWERS' COMMENTS

Reviewer #1 (Remarks to the Author):

The addition of the scRNA-Seq data greatly improves the characterization of the fetal and adult organoids. The text improvements were helpful for improving the clarity of this paper, and we thank the authors for clarifying the effect of FSK on hepatocytes vs cholangiocyte cultures. There is one minor area that need to be addressed: the clarifications provided about the different types of cultures were helpful. However, it would be helpful to indicate whether the results obtained from the 4 different donors were consistent.

intentionally signed: Gary Peltz

Reviewer #2 (Remarks to the Author):

My comments on the first version have been addressed adequately and I can recommend publication.

Reviewer #1 (Remarks to the Author):

The addition of the scRNA-Seq data greatly improves the characterization of the fetal and adult organoids. The text improvements were helpful for improving the clarity of this paper, and we thank the authors for clarifying the effect of FSK on hepatocytes vs cholangiocyte cultures. There is one minor area that need to be addressed: the clarifications provided about the different types of cultures were helpful. However, it would be helpful to indicate whether the results obtained from the 4 different donors were consistent.

intentionally signed: Gary Peltz

A: Thank you. We have now indicated when which donor was used in each experiment in the Statistics and Reproducibility section in the Methods. All 4 fetal donors grew into typical human fetal hepatocyte organoid cultures and were phenotypically characterized. All bulk transcriptomic and phenotypic experiments were performed using 2 donors (varying in combinations across the total 4 donors), showing consistent results. Human fetal hepatocyte organoids from 1 donor were used for single-cell RNA sequencing.

Reviewer #2 (Remarks to the Author):

My comments on the first version have been addressed adequately and I can recommend publication.

A: Thank you.